:ᬑ: PLOS | ONE

# Sequencing artifacts derived from a library preparation method using enzymatic fragmentation

**Norio Tanaka[1][◉], Akihisa Takahara[1,2][◉], Taichi Hagio[1,2], Rika Nishiko[1], Junko Kanayama[1], Osamu Gotoh[1], Seiichi Mori[1]***

**1** Project for Development of Innovative Research on Cancer Therapeutics, Cancer Precision Medicine Center, Japanese Foundation for Cancer Research, Ariake, Koto-ku, Tokyo, Japan, **2** Data4C's Co. Ltd., Minami-azabu, Minato-ku, Tokyo, Japan

◉ These authors contributed equally to this work.

\* seiichi.mori@jfcr.or.jp

**Data Availability Statement:** The two rectal cancer datasets are available on National Bioscience Database Center (NBDC; https://humandbs.biosciencedbc.jp/) with Accession ID hum0140.

## Abstract

DNA fragmentation is a fundamental step during library preparation in hybridization capture-based, short-read sequencing. Ultra-sonication has been used thus far to prepare DNA of an appropriate size, but this method is associated with a considerable loss of DNA sample. More recently, studies have employed library preparation methods that rely on enzymatic fragmentation with DNA endonucleases to minimize DNA loss, particularly in nano-quantity samples. Yet, despite their wide use, the effect of enzymatic fragmentation on the resultant sequences has not been carefully assessed. Here, we used pairwise comparisons of somatic variants of the same tumor DNA samples prepared using ultrasonic and enzymatic fragmentation methods. Our analysis revealed a substantially larger number of recurrent artifactual SNVs/indels in endonuclease-treated libraries as compared with those created through ultrasonication. These artifacts were marked by palindromic structure in the genomic context, positional bias in sequenced reads, and multi-nucleotide substitutions. Taking advantage of these distinctive features, we developed a filtering algorithm to distinguish genuine somatic mutations from artifactual noise with high specificity and sensitivity. Noise cancelling recovered the composition of the mutational signatures in the tumor samples. Thus, we provide an informatics algorithm as a solution to the sequencing errors produced as a consequence of endonuclease-mediated fragmentation, highlighted for the first time in this study.

## Introduction

Next-generation sequencing (NGS) technologies have facilitated the delivery of precision medical care to patients with cancer. Short-read sequencing technology has been widely exploited for this purpose, encompassing amplicon- or hybridization capture-based library preparations [1]. This diagnostic strategy relies on accurate sequencing and interpretation to provide patients with the right clinical decision [1, 2]. Genome sequencing using hybridization capture

The other data (lung and two gastric cohorts) are not publicly accessible because our institutional review board prohibits public sharing of the data. However, these data can be available upon request to the JFCR Data Control Office (Data_Control@jfcr.or.jp).

**Funding:** MEXT | Japan Society for the Promotion of Science (JSPS) - JP15K06861 [S. M. ]; MEXT | Japan Society for the Promotion of Science (JSPS) - JP17K18337 [O. G. ]; MEXT | Japan Society for the Promotion of Science (JSPS) - JP18K07338 [S. M. ]. This work was supported by JSPS KAKENHI Grant Numbers JP17K18337, JP15K06861, and JP18K07338. Data4C's co. ltd. was neither funding nor funded by any governmental research funding agency. The funder had no role in study design, data collection and analysis, decision to publish, or preparation of the manuscript. Whereas Data4C's co. ltd. provided support in the form of salaries for authors A.T. and T.H., Japanese Foundation for Cancer Research paid to Data4C's co. ltd. as a business consignment for their assisting analytic parts of this work.

**Competing interests:** A.T. and T. H. are employed by Data4C's. co. ltd. This does not alter our adherence to PLOS ONE policies on sharing data and materials. There are no patents or products in development or marketed products associated with this research to declare.

comprises multiple steps, including tissue processing, tissue storage, DNA isolation, DNA fragmentation, probe hybridization, library amplification, sequencing, and informatics analysis [2–5]. Sequencing errors can be introduced at any of these steps, and nucleotides can be further modified through oxidation during tissue processing, tissue storage, DNA isolation, and DNA fragmentation [2–4, 6]. Nucleotide incorporation errors can in turn create polymerase reaction biases, affect precise library amplification, and generate sequencing noise [2–4]. Although substantial efforts have been made to minimize such sequencing noise experimentally, stochastic errors remain persistent [2–4].

Ultra-sonication has long been a standard method for DNA fragmentation during hybridization capture-based, short-read sequencing. Ultrasonication creates even cuts in the DNA across the entire genome, thereby providing a simple means of controlling fragment size in a non-biased manner [2, 3]. However, the physical scattering of DNA solution during the process often leads to a loss of DNA sample, which can be critical when the sample amount is limited to nano- or picogram quantities, such as found with biopsied tissue fragments. Several commercial library preparation kits are available, including the HyperPlus (KAPA Biosystems), SureSelect QXT (Agilent Technologies), Fragmentase (New England Biolabs), and Nextera Tagmentation (Illumina) kits, each of which uses endonucleases or transposases for DNA fragmentation. Although these kits minimize DNA loss, it remains largely unknown what degree of sequencing errors are caused by the enzymatic fragmentation process.

In the current study, we identified numerous artifactual SNVs/indels among libraries constructed using the HyperPlus kit for DNA fragmentation. These sequencing errors—characterized by variants located at the center of palindromic structures and near the 5' or 3' ends of the read, with multi-nucleotide substitutions—were deemed to have been introduced by the endonuclease treatment step and the following fill-in process for end repair, but not as a result of the whole sequencing process *per se*. Taking advantage of these noise properties, we developed an algorithm to efficiently distinguish sequencing errors from genuine mutations. This algorithm could be used in future studies to improve datasets that rely on enzymatic fragmentation using the same or a similar enzyme during library preparation.

## Results

### Distinct features of somatic SNVs/indels derived from different library preparation kits

In our sequencing facility, the SureSelect kit (Agilent, Santa Clara, California, United States) is the default protocol for library preparation for exome analysis and requires 200 ng of DNA as the starting amount. However, in some cases, the starting amount is less than 200 ng; this typically occurs when samples are extracted from small tissue fragments. For such samples, we use the HyperPlus kit (KAPA Biosystems, Cape Town, South Africa) for library preparation, which requires a minimum of only 20 ng of DNA. Typically, there tends to be sufficient amount of matched normal DNA for the standard processing, and consequently—although not ideal—it is often the case that somatic mutation calling occurs for tumor and matched normal samples prepared using different DNA fragmentation methods.

Using exome sequencing of tumor samples—with the exception of hypermutators—we will typically detect several tens or hundreds of somatic SNVs/indels per sample. However, in our experience, we noted that some tumor samples exhibited an extraordinarily large number of SNVs/indels, exceeding a few thousand, regardless of tissue origin, histological type, or method of tissue preservation. We also noted that these tumor samples were prepared using the HyperPlus kit and that the paired normal DNA samples were prepared with the SureSelect kit. Further inspection of data pertaining to 31 tumors (16 gastric, 13 lung and 2 rectal cancers)

prepared using the HyperPlus kit revealed a higher median number of SNVs (median: 2,308, range: 1,119–3,996) and indels (median: 89, range: 33–437) as compared with data from tumor samples prepared using the SureSelect kit.

This obvious discrepancy prompted us to perform pairwise comparisons of sequenced reads from the same tumor DNA libraries prepared with the SureSelect kit versus those prepared with the HyperPlus kit (Fig 1). Six tumor tissues preserved as fresh-frozen samples were used for this analysis. For somatic calls, normal DNA samples were prepared using the SureSelect kit. In our comparisons of the exome sequencing process, we noted one major difference between the two library preparation kits: the SureSelect kit uses ultra-sonication for DNA fragmentation whereas the HyperPlus kit relies on endonuclease treatment. Quality assessment of the sequencing data from the six tumor samples generated by both methods showed no difference in the percentage of reads with Q30 values (the Phred score assigns a Q score of 30). However, the percentage of on-target reads, which is dependent on the library preparation method, differed between the SureSelect and HyperPlus kits (Table 1). Therefore, we obtained and

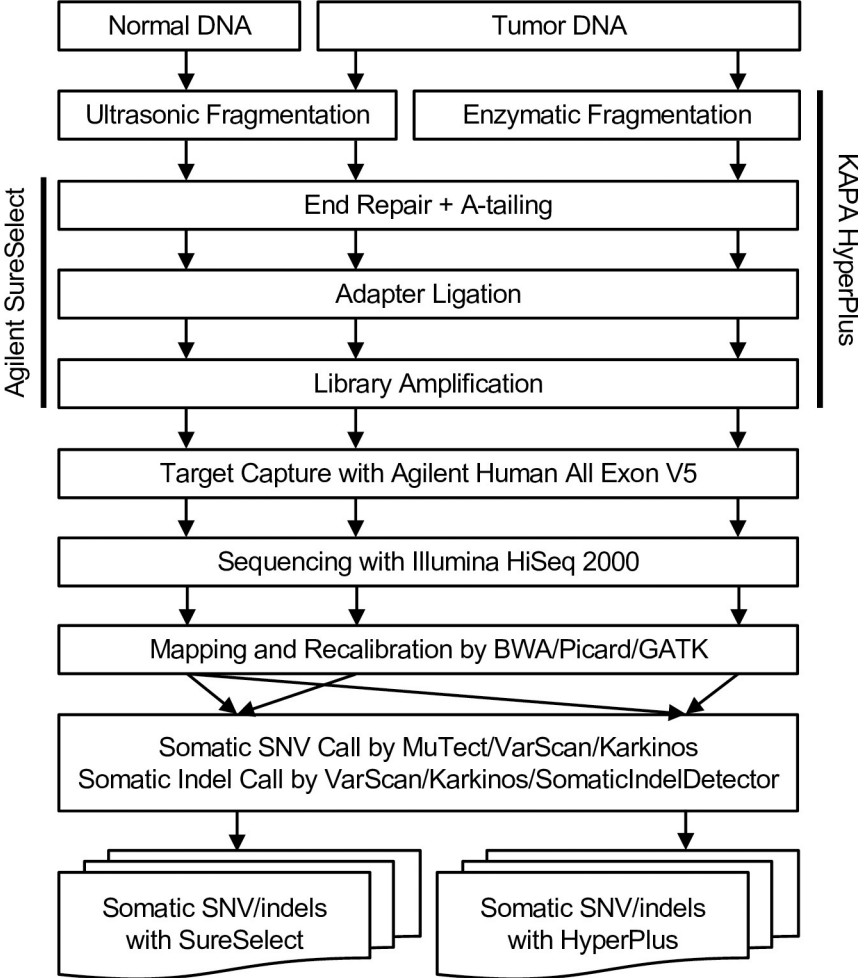

**Fig 1. Experimental procedure to prepare sequencing libraries using Agilent SureSelect or KAPA HyperPlus kits, and the analytical pipeline to call somatic SNVs and indels.** A major difference between the two kits is the use of ultra-sonication versus enzymatic treatment with an endonuclease for DNA fragmentation. The same tumor DNA samples were used for library preparation, and the SureSelect kit was used for all of the paired normal DNA in the somatic mutation detection.

**Table 1. Quality assessment of sequencing.**

| Sample | Study Cohort | Number of Mapped Reads | | On-Target Reads (%) | | Reads ≥ Q30 (%) | |
|---|---|---|---|---|---|---|---|
| | | HyperPlus | SureSelect | HyperPlus | SureSelect | HyperPlus | SureSelect |
| A001 | Lung Cancer | 477,599,892 | 437,724,129 | 67.04% | 74.85% | 94.43 | 95.46 |
| A004 | | 512,922,172 | 436,510,390 | 55.35% | 73.67% | 93.68 | 95.42 |
| A005 | | 520,804,987 | 455,109,465 | 56.65% | 74.93% | 93.13 | 94.91 |
| B012 | | 496,356,981 | 433,005,359 | 57.48% | 74.58% | 92.39 | 95.39 |
| C742 | Rectal Cancer | 198,466,488 | 359,482,010 | 61.26% | 71.63% | 93.99 | 95.93 |
| C772 | | 230,911,546 | 362,714,354 | 58.19% | 71.67% | 94.33 | 95.94 |

analyzed somatic SNVs and indels from two combinations: tumor DNA–SureSelect with normal DNA–SureSelect ("Somatic SNV/indels with SureSelect") and tumor DNA–HyperPlus with normal DNA–SureSelect ("Somatic SNVs/indels with HyperPlus") (Fig 1).

The features of the somatic SNV/indel call results derived from the SureSelect and HyperPlus treatments are shown in Fig 2. Despite starting with the same sample of DNA, the HyperPlus libraries resulted in 2.3- to 9.9-times more SNV/indel detections than the SureSelect libraries (Fig 2A). Importantly, most of SNVs/indels derived from the SureSelect treatment were nested within those from the HyperPlus libraries, but most of the SNVs/indels from the HyperPlus treatment were not also common to the SureSelect-treated libraries (Fig 3A). Given that the numbers of SNVs/indels from the SureSelect libraries for the six tumor samples were comparable with that described in previous literature [7, 8], we concluded that the HyperPlus libraries generated a substantial number of somatic SNVs/indels as non-biological sequencing artifacts.

Closer inspection of the data uncovered that many of these somatic SNVs were coincidently located at the center of palindromic sequences, herein designated as "SNV-centered palindromes" (SCPs). HyperPlus libraries also more frequently generated longer SCPs, whereas no SCP over 15 bases in length was detected among the SureSelect libraries (Fig 2B).

COSMIC mutational signature analyses [10] were performed to assess the overall pattern of the SNV artifacts generated by the HyperPlus kit; the scores are depicted as heatmaps in Fig 2C. Consistent with previous reports [11, 12], the SureSelect libraries produced tumor-type–associated signature profiles, with higher scores for signatures 1, 2, and 4 for lung cancers (A001, A004, A005 and B012) and signatures 1 and 6 for rectal cancers (C742 and C772). However, the HyperPlus libraries showed constant peaks associated with signatures 3, 4, and 22, even across tissues of origin, indicating that the HyperPlus treatment generated a specific set of nucleotide substitutions in the genome as the "HyperPlus signature" (Fig 2C). From these observations, we concluded that the HyperPlus treatment method led to non-biological sequencing artifacts, with biased nucleotide substitutions at characteristic palindromic parts of the genome.

## Attributes of sequencing artifacts by HyperPlus

To obtain a more detailed characterization of these sequencing artifacts, we divided the somatic SNV/indel calls from the HyperPlus libraries into two categories of variants: HyperPlus-specific SNVs/indels (category [a]), and commonly detected SNVs/indels, which were found with both libraries (category [b]). This categorization was based on the premise that most of the SNVs/indels in category [a] were likely to be noise generated by the HyperPlus method of preparation, and that genuine somatic SNVs/indels would predominantly be found in category [b] (Fig 3A). We noted three distinctions in the SNVs/indels between the two

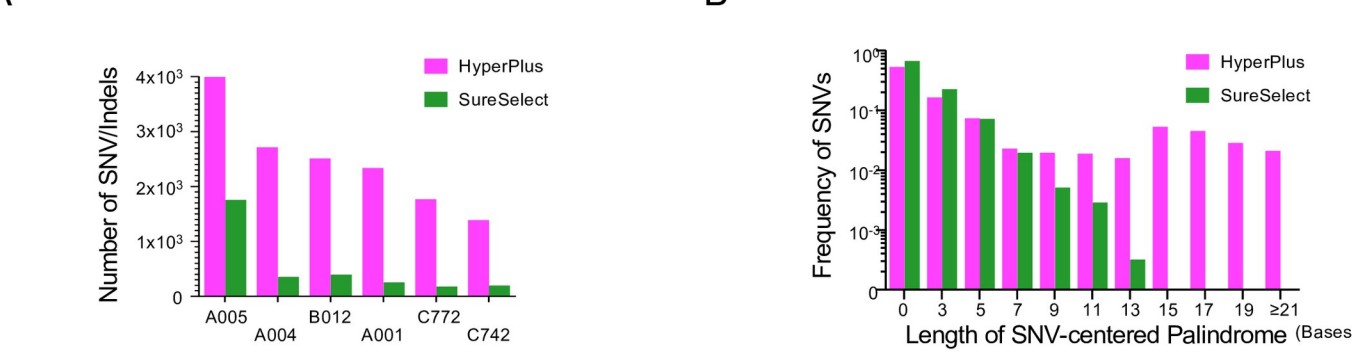

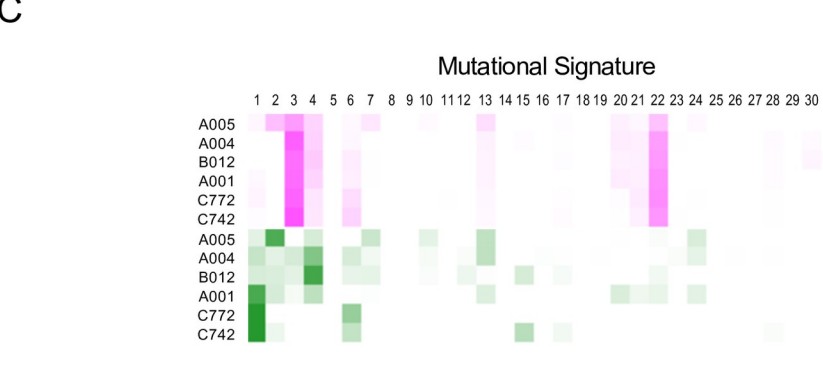

**Fig 2. Differences in the sequenced results using the HyperPlus kit versus the SureSelect kit.** For each sample, the same DNA was sequenced with the two different kits: HyperPlus (magenta) and SureSelect (green) kits. (A) Number of SNVs/indels. Bar plots show the number of SNVs and indels. (B) Frequency of SNVs per length of palindrome. The frequency of SNVs from all samples ($n = 6$) is shown as bar plots per odd length of palindrome, in which a somatic SNV was detected at the center. The number on the $x$-axis indicates the total length of the palindrome. Note that a length "zero" indicates a sequence that lacks a palindrome. (C) Mutational signature heatmap. The heatmap represents the proportions of the 30 COSMIC mutational signatures [9] computed from trinucleotide frequencies of nucleotide substitutions in each sample. The number above the heatmap indicates each of the 30 signatures computed according to the definitions in the COSMIC database. Sample IDs are shown on the left.

categories: 1) Most SNVs/indels in category [a] were at least once detected across the pooled data from the HyperPlus libraries, but this was not the case for those in category [b] (Fig 3B); 2) SNVs/indels in category [a] were frequently located 10- to 15-bases away from the 5' or 3' edge of the read (defined as "positional bias"), whereas SNVs/indels in category [b] were more uniformly distributed (Fig 3C); 3) Reads with SNVs/indels from category [a] were more substantially soft-clipped than those from category [b] (50.8% and 5.0% on average), which implies multi-nucleotide substitutions at the 5' or 3' end of the read (Fig 3D).

## Designing a filtering algorithm to remove sequencing artifacts derived from HyperPlus

Despite these shortcomings, enzymatic fragmentation for library preparation is often unavoidable, particularly when only nanograms of DNA sample is available. Taking advantage of the

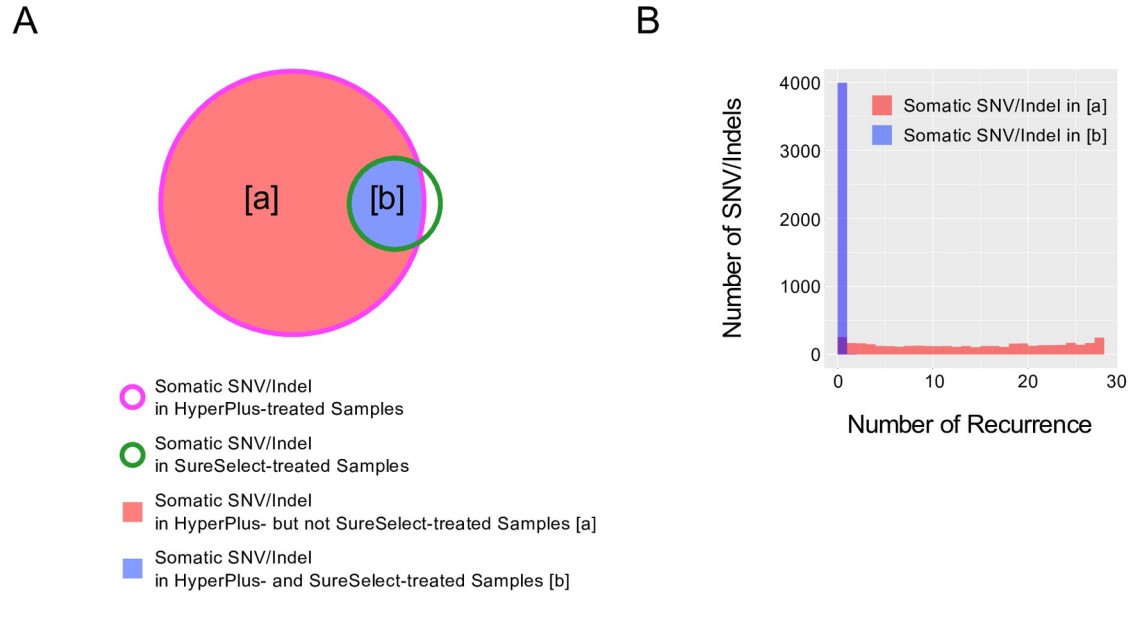

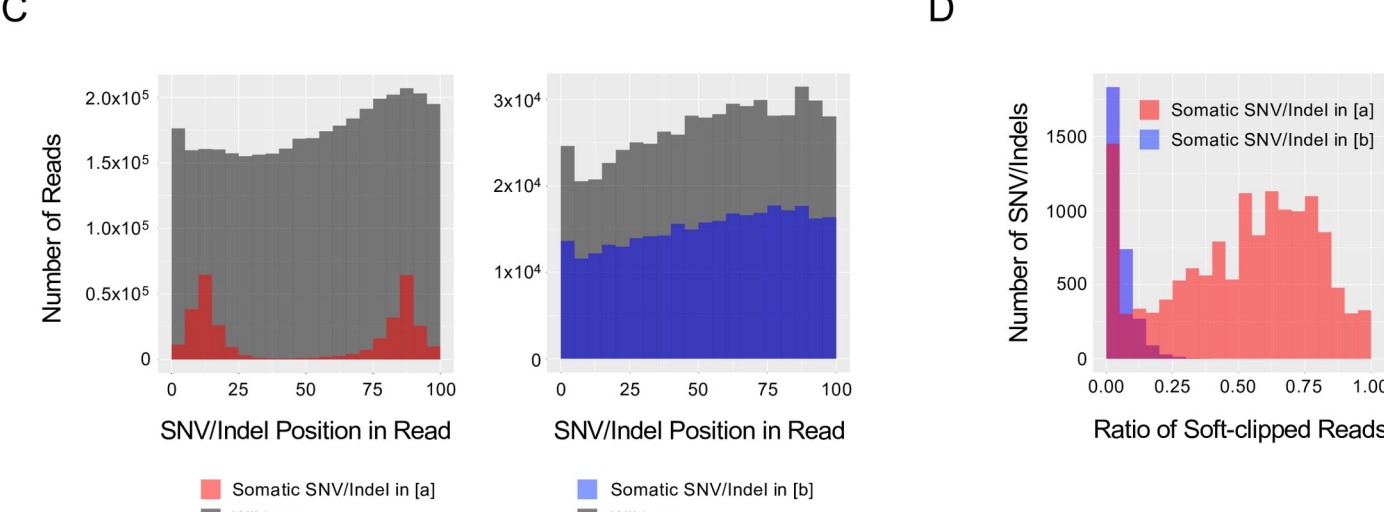

**Fig 3. Features of SNV/indel noise caused by the HyperPlus kit.** (A) Venn diagram of somatic SNVs/indels detected in the HyperPlus- and SureSelect-treated samples. Magenta and green lines indicate HyperPlus and SureSelect treatments, respectively. The regions [a] (red) and [b] (blue) indicate somatic SNVs/indels detected in the HyperPlus-treated but not the SureSelect-treated samples (red), and those shared by both HyperPlus- and SureSelect-treated samples (blue), respectively. Note: features were extracted from BAM files from libraries prepared with the HyperPlus kit for tumor samples. (B) Histogram depicting recurrence in somatic SNV/indel detection across 28-sample libraries prepared with the HyperPlus kit. *X*- and *y*-axes indicate the number of recurrent detections of identical SNVs/indels and the number of SNV/indels, respectively. Red and blue colors indicate somatic SNVs/indels in [a] and [b] in (A), respectively. (C) Histogram for the distribution of SNV/indel position in the read. *X*- and y-axes indicate the positions of SNVs/indels within 5-bases and the number of reads, respectively. Left panel. Position of somatic SNVs/indels detected in [a] (red). Right panel. Positions of somatic SNVs/indels detected in [b] (blue). The number of wildtype nucleotide reads that mapped to the same genomic coordinate as the detected SNV/indel is indicated in grey. (D) Histogram for the ratio of the soft-clipped reads. *X*- and *y*-axes indicate the ratio of soft-clipped reads at 0.05 intervals and the number of SNV/indels, respectively.

salient properties of the sequencing noise generated by the HyperPlus method, we sought to develop a filtering algorithm to remove these artifacts from somatic SNV/indel call results to optimize the sequencing data (Fig 4A). The algorithm comprised two filtering steps: First, we excluded recurrently detected SNVs/indels across the pooled HyperPlus data, unless the

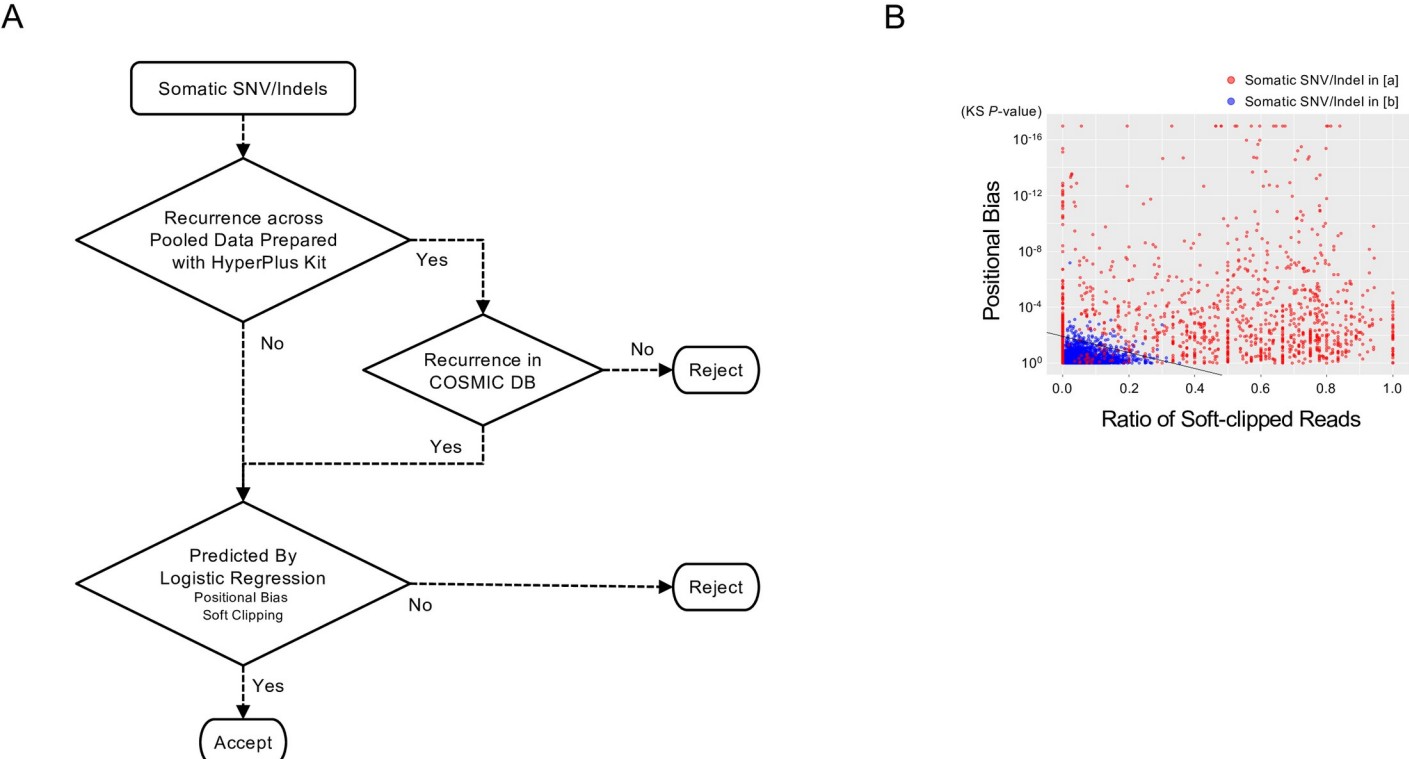

**Fig 4. Filtering process for the removal of somatic SNV/indel noise generated by treatment with the HyperPlus kit.** (A) Algorithm for the filtration of somatic SNV/indel noise in libraries prepared with the HyperPlus kit. (B) Relationship between soft clipping of a read and an SNV/indel position in the read. 2-D scatter plots present the relation between the ratio of soft-clipped reads and the *p*-value, as determined using the Kolmogorov-Smirnov (KS) test for positional bias of the SNV/indel in the read. Circles denote SNVs/indels detected using the HyperPlus kit only (red; category [a] in Fig 3A) and SNVs/indels detected using both the HyperPlus and SureSelect kits (blue; category [b] in Fig 3A). The black line indicates the threshold by logistic regression to distinguish SNVs/indels in category [a] from those in category [b] according to the generalized linear model.

SNVs/indels were already registered in the COSMIC database. Second, we developed and utilized a predictive model to remove SNVs/indels that showed a positional bias in a read and/or those on frequently soft-clipped reads. Positional bias was quantified using the Kolmogorov-Smirnov (KS) test to compare variant and wildtype alleles. The extent of soft clipping was measured using the ratio of soft-clipped reads per total reads with SNVs/indels. Overall, the predictive model was based on logistic regression to classify the SNV/indel as noise or signal (Fig 4A).

2-D scatter plots show the relationship between the KS *p*-value for positional bias and the ratio of soft-clipped reads (Fig 4B). Whereas category [a] SNVs/indels (mostly sequencing artifacts) were characterized by lower KS *p*-values and/or a higher ratio of soft-clipped reads, SNVs/indels in category [b] (mostly genuine SNVs/indels) had higher KS *p*-values and a lower ratio of soft-clipped reads. A threshold was then estimated to distinguish SNVs/indels between the two categories using a generalized linear model with the logit link function. Using receiver operating characteristic (ROC) curve analysis for the six-sample training data, the final model was established and shown to be capable of distinguishing SNVs/indels between the two categories with a specificity of 0.914 and a sensitivity of 0.979 (Fig 4B).

## Noise reduction in the training data

We next applied our noise filtering algorithm to the six-sample training data to assess how filtering affects the data derived using the HyperPlus and SureSelect treatments (Fig 5). The total

numbers of SNVs/indels in category [a] (likely noises from HyperPlus treatment) and category [b] (likely genuine mutations) were 11,731 and 2,984, respectively. Of these, recurrently detected SNVs/indels across the in-house pooled data prepared with the HyperPlus kit reached 10,928 and 16, of which 389 (3.6%) and 11 (68.8%) were registered in the COSMIC database (ver. 82). Because these 389 and 11 SNVs/indels were considered probably genuine, they were returned to the filtering process. This left 1,192 and 2,979 SNVs/indels in categories [a] and [b], respectively.

We then proceeded to the next step of the logistic regression based on positional bias and soft clipping. The predictive model classified 1,090 and 62 SNVs/indels as HyperPlus noise in categories [a] and [b], respectively. As anticipated, after filtering, most of the SNVs/indels in category [a] (99.1%; 11,628/11,731) were removed, but far fewer SNVs/indels were removed for category [b] (2.2%; 67/2,984) (Fig 5A and 5B). The resultant number of SNVs/indels in the HyperPlus data after filtering was similar to that in the unfiltered SureSelect data (Fig 5E, left panel). Filtering efficiently removed SNVs with SCPs longer than 13 nucleotides from category [a], whereas most of the SNVs in category [b] remained in the group (Fig 5C). Among 11,695 filtered SNVs, 3,407 SNVs were located at the center of odd-length palindromes (length ≥ 5 bases) and 66.6% of such SCPs were recurrently observed across the samples (Fig 5D). An inspection of the substrings of the palindromes revealed substantial diversity in the length and nucleotide sequence among the samples (371–655 [median 568] different palindromes per sample; Fig 5D and S1 Table). Furthermore, consistent with the presence of positional bias of the artifactual SNVs (Fig 3C), we found that, in 90.4% of SCPs, the entire palindrome sequence was nested within 30 bases from the edge of the read (S1 Fig).

Consequently, the frequency of SNVs per length of SCP among the HyperPlus data after filtering was normalized to that of the SureSelect data (Fig 5E middle panel). Similarly, filtering rendered the mutational signature profiles of the six tumors mostly indistinguishable between the HyperPlus and SureSelect treatments (Fig 5E right panel). These observations confirmed the validity of the filtering algorithm in the six-tumor training samples.

### Noise reduction in test data

We next assessed the effects of the filtering algorithm on the remaining samples not used to develop the predictive model for filtering (Fig 6). For this, we used 39 tumor data derived from three independent genomic cohorts: a gastric cancer cohort 1 (*n* = 3), a lung cancer cohort (*n* = 9), and a gastric cancer cohort 2 (*n* = 27). Among the 39 samples, 25, 9, and 5 samples were sequenced with the KAPA HyperPlus, KAPA Hyper, and Agilent SureSelect library preparation kits, respectively. There were nine formalin-fixed paraffin-embedded (FFPE) and 30 fresh-frozen tumor samples. We show the number of SNVs and indels, and the pattern of the mutational signatures before and after filtering the data (Fig 6).

The experimental procedure for the Hyper kit is similar to that for the HyperPlus kit, except that the Hyper kit uses ultra-sonication for DNA fragmentation instead of endonuclease treatment, similar to the SureSelect kit. Noteworthy, there was no significant difference in the number of SNVs/indels or the pattern of the mutational signature between the SureSelect-treated and Hyper-treated samples before filtering, suggesting that the Hyper kit *per se* does not produce the sequencing errors recorded for the HyperPlus kit.

Filtering substantially reduced the number of SNVs/indels in the HyperPlus data but had little effect on the Hyper and SureSelect data. The median (range) proportions of the remaining SNVs/indels were 10.8% (0.01%–46.9%), 85.2% (47.6%–98.8%), and 94.3% (86.5%–98.6%) for the HyperPlus, Hyper, and SureSelect datasets, respectively (Fig 6A). In the mutational signatures (Fig 6B), filtering removed cancer type-independent peaks for signatures 3, 4, and 22,

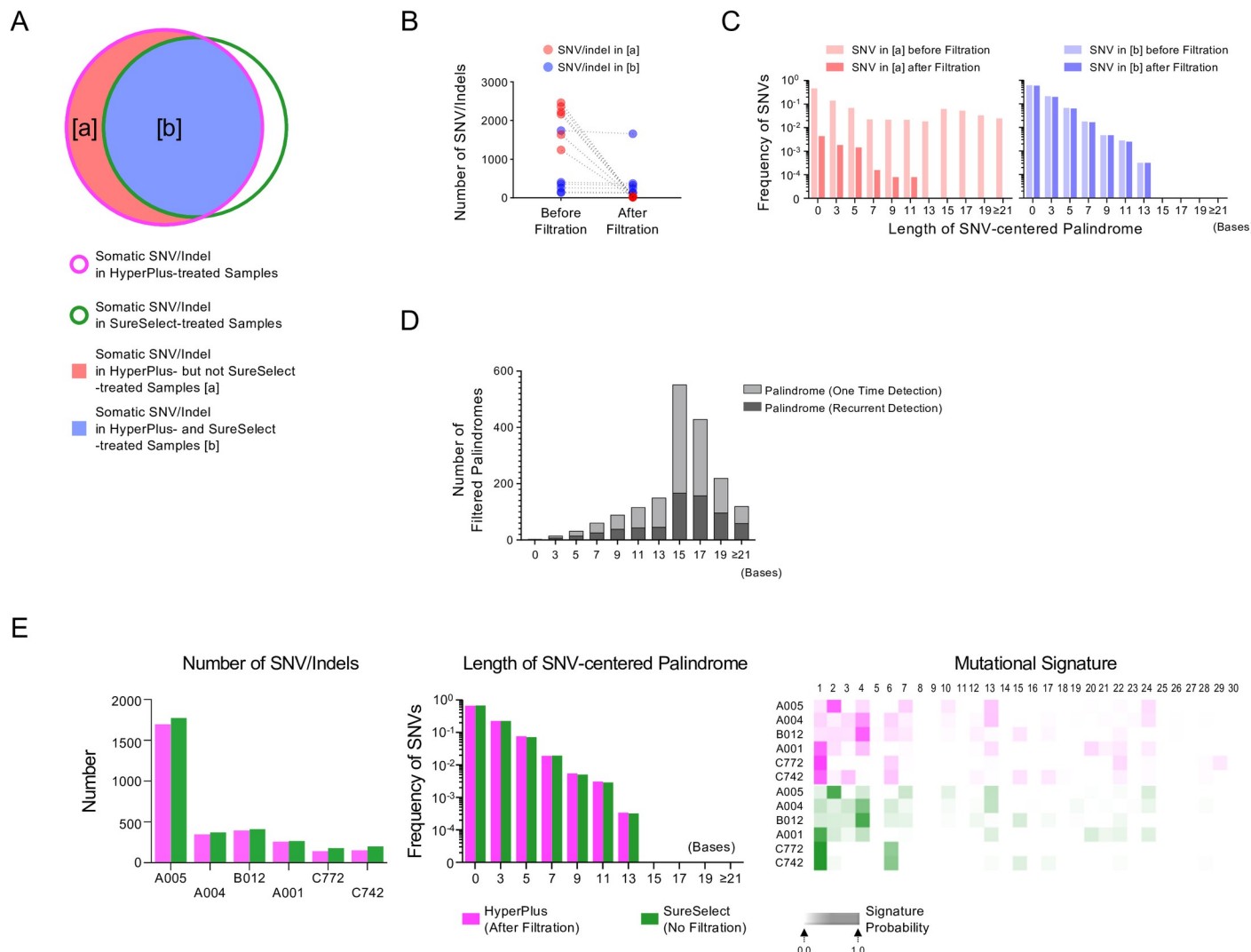

**Fig 5. Noise reduction in the six-sample training data.** We show changes in the features before and after noise filtration for each sample. (A) Venn diagram of somatic SNVs/indels in the HyperPlus- and SureSelect-treated samples after noise filtering. Magenta and green lines indicate HyperPlus and SureSelect treatments. Region [a] (red) indicates somatic SNVs/indels detected in the HyperPlus- but not the SureSelect-treated samples, whereas region [b] (blue) indicates those shared by both treated samples. The features shown in (A) and (B) were extracted from BAM files from the HyperPlus libraries. (B) Dot plots for changes in the SNV/indel number for each sample after noise reduction. (C) Frequency of SNVs per length of palindrome before and after filtering. Categories [a] and [b] are shown on the left and right panels. Faded and full shading indicate before and after filtration. The frequency of SNVs from all six samples is shown as bar plots per odd length of palindrome, in which a somatic SNV was detected at the center (SCP: SNV-centered palindrome). The number on the x-axis indicates the total length of the palindrome. A length "zero" indicates a sequence that lacks a palindrome. (D) Number of filtered palindromes in a histogram. The number of SCPs was counted per odd-number-length palindrome and shown. Gray and black colors indicate one time and recurrently detected palindromes across the samples, respectively. The reference allele sequence was used to count the SCPs. (E) Comparisons of several features in the sequenced results between the HyperPlus and SureSelect treatments after filtering the HyperPlus artifacts. Note that the noise filter was applied to the HyperPlus data but not to the SureSelect data. Left panel: Bar plots for number of SNVs/indels. Middle panel: Frequency of SNVs per length of palindrome. The frequency of SNVs from all samples (n = 6) is shown as bar plots per odd length of palindrome, in which a somatic SNV was detected at the center. The number on the x-axis indicates the total length of the palindrome. Note that a length "zero" indicates a sequence that lacks a palindrome. Right panel: Mutational signature heatmap. The heatmap represents the proportions of 30 COSMIC mutational signatures [9] computed from trinucleotide frequencies of nucleotide substitutions in each sample. The number above the heatmap indicates each signature computed according to the definitions in the COSMIC database. Sample IDs are shown on the left.

with a more uniform distribution of signatures in the HyperPlus data. Subsequently, the noise reduction rendered more signals for signatures 1 and 6 for both gastric cancer cohorts (#1 and #2) and signatures 1, 4, 7, 13, 20, 22, and 24 for the lung cancer cohort (Fig 6B). On the other hand, the filtering algorithm did not change the profiles of the Hyper and SureSelect data (Fig

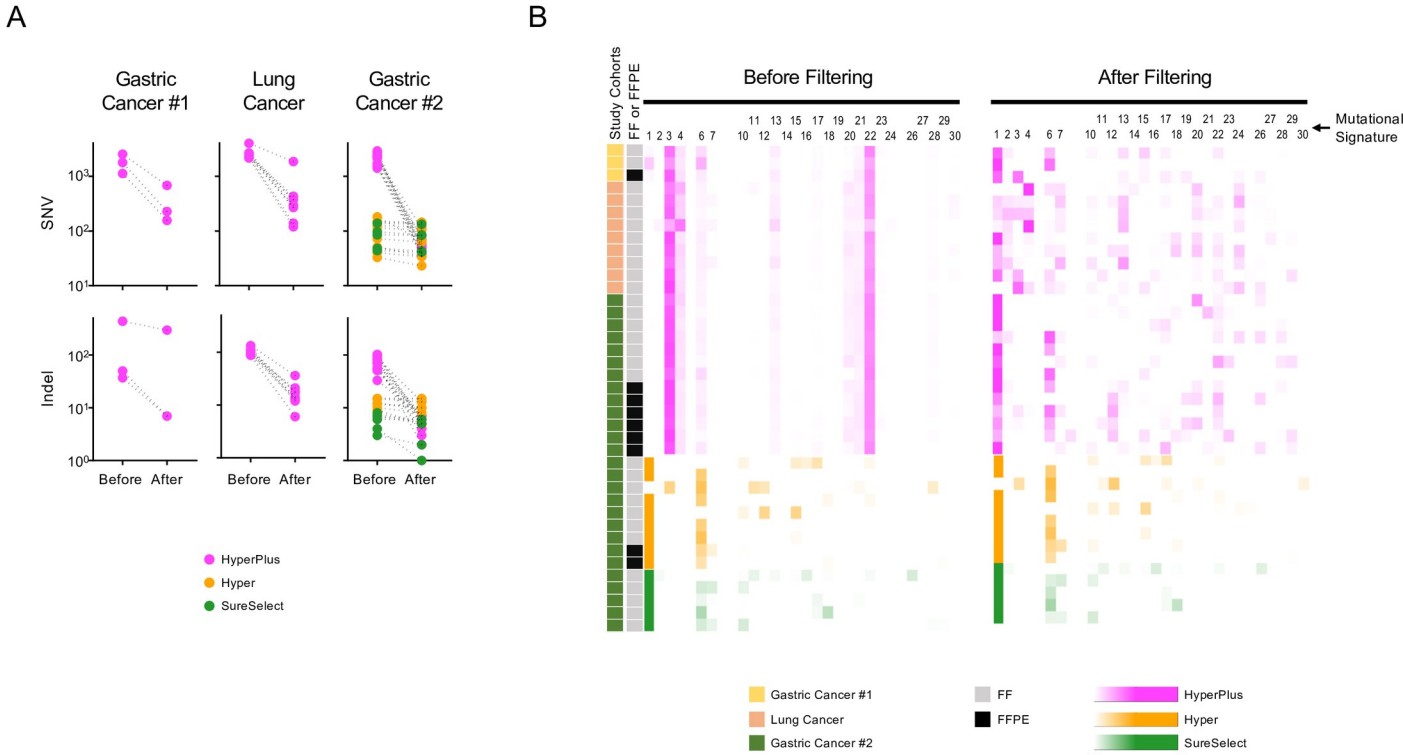

**Fig 6. Noise reduction in the test data.** Changes in the features of the somatic variants detected before and after noise filtration for each sample. Libraries were prepared with KAPA HyperPlus (enzymatic fragmentation), KAPA Hyper (ultrasonic fragmentation), or Agilent SureSelect (ultrasonic fragmentation) kits. Note that the only difference between the HyperPlus and Hyper kits is that the Hyper kit uses ultra-sonication in the DNA fragmentation step, whereas endonuclease treatment is used in the HyperPlus kit. (A) Dot plots show changes in the numbers of SNVs (upper panels) and indels (lower panels) for each sample, with libraries prepared with HyperPlus (magenta), Hyper (orange), or SureSelect (green) kits. (B) Heatmap presentations of mutational signatures before and after noise filtration for each sample. Libraries were prepared with HyperPlus (magenta), Hyper (orange), or SureSelect (green) kits. The heatmap represents the proportions of the 30 COSMIC mutational signatures [9] computed from trinucleotide frequencies of nucleotide substitutions in each sample. The number above the heatmap indicates each of the 30 signatures computed according to the definitions in the COSMIC database. The study cohorts and tissue preservation types are indicated on the left.

6B). These observations demonstrate that the artifacts introduced by sample preparation with the HyperPlus kit were removed selectively and efficiently by the noise reduction algorithm.

## Reduced but persistent artifactual SNVs/indels by somatic mutation calling with normal–HyperPlus/ tumor–HyperPlus libraries

For a more controlled analysis, we sought to examine the error production rates with the HyperPlus and SureSelect kits using the same DNA fragmentation method in paired normal–tumor samples; e.g., HH combination (normal–HyperPlus versus tumor–HyperPlus) and SS combination (normal–SureSelect versus tumor–SureSelect). Normal and tumor samples from two rectal cancer cases (C742 and C772) were analyzed. We found a substantially reduced number of somatic SNVs/indels for the HH combination (190 and 168 for C742 and C772 tumors), which was almost similar to that found for the SS combination (194 and 179 for C742 and C772 tumors). Whereas the SS and HH combinations detected common and specific SNVs/indels, it is important to note that the HyperPlus-associated sequencing errors and the production of SNV-centered palindromes were persistent in the HH combination, as detected by the filtering algorithm (Table 2). These findings clearly indicate an experimental difficulty in being able to completely cancel the sequencing noise produced by the HyperPlus treatment,

**Table 2. Somatic mutation calling with the same DNA fragmentation method in pairs of normal and tumor DNA.**

| | HH and SS Common | | HH Only | | SS Only | |
|---|---|---|---|---|---|---|
| | C742 | C772 | C742 | C772 | C742 | C772 |
| No. of Detected SNVs/Indels | 165 | 140 | 25 | 28 | 29 | 39 |
| No. of SNVs/indels Classified as HyperPlus Noise | 3 | 5 | 17 | 21 | 6 | 2 |
| No. of SNVs with SCP Length ≥15 | 0 | 0 | 2 | 6 | 0 | 0 |

Abbreviations:

HH: Normal DNA treated with HyperPlus kit/ Tumor DNA treated with HyperPlus kit

SS: Normal DNA treated with SureSelect kit/ Tumor DNA treated with SureSelect kit

SCP: SNV-Centered Palindrome

even after using the same fragmentation method in paired normal and tumor samples, and suggest the necessity of using informatics to filter the noise.

## Discussion

The advent of NGS has meant that DNA analysis can be achieved in an efficient and highly sensitive high-throughput manner, offering a means to generate large amounts of data, decipher the subtle yet potentially informative distinctions between samples, and help to facilitate an understanding of genetic disease. In hybridization capture-based short-read sequencing, DNA fragmentation is a necessary step in the preparation of nucleic acids, as the quality of the sequencing is contingent on both the randomness of the DNA fragmentation as well as the overlap of the resultant fragments. Furthermore, because fragment size tends to differ across NGS platforms and sequencing runs, efficient control of DNA fragment size is imperative.

Ultra-sonication is one such method that can control DNA fragment size by evenly cleaving DNA throughout the entire genome and, as such, has remained a gold standard in sequencing. However, studies have reported that ultra-sonication produces sequencing noise in the form of oxidative nucleotide modifications, such as guanine to 8-oxo guanine (8-oxo-G) and cytosine deamination [2, 6, 13]. Nebulization is another commonly used mechanical method of DNA shearing. In this method, compressed nitrogen or air is forced into the DNA through a small hole, generating random sheared fragments with both overhangs and blunt ends.

In addition to these mechanical modes of fragmentation, several kits have been developed recently using enzymatic treatment to shear the DNA; albeit, it remains largely unknown whether sequencing errors occur with these alternative modes of cutting. One previous report showed that Fragmentase (New England Biolabs) causes more artifactual indels than sonication or nebulization; although, the number of indels generated by Fragmentase appeared to be within the two-fold range of that produced by the physical methods [14].

We consider the sequencing noise in the HyperPlus-treated samples to be derived as a consequence of endonuclease treatment. There are three major reasons for this proposition. First, we note positional biases in the mutations, with errors frequently located 10- to 15-bases from the 5' or 3' end of the read. This implies that the positions are associated with the cutting sites of the HyperPlus endonuclease. Second, the Hyper kit, manufactured by the same company as the HyperPlus kit, uses ultra-sonication for DNA fragmentation instead of endonuclease treatment, and did not produce the same amount of noise as that generated by the HyperPlus kit. Third, artifactual SNVs were often observed at the centers of palindromic sequences, suggestive of another bias in sequence recognition by the endonuclease(s) in the fragmentation step.

Previous studies have highlighted biases in the cleavage sites targeted by "non-specific" endonucleases, such as DNase I [15–18]. The HyperPlus endonuclease—the type and

composition have not been disclosed (KAPA Biosystems)—seemingly has preferential recognition sites for genomic DNA, and these include palindromic sequences. Importantly, the SCPs were not only substantially diverse in length and sequence but also 66.6% of SCPs recurrently appeared across a range of samples. In addition, in almost all (90.4%) of the SCPs, the entire palindromic sequence was nested within 30 bases from the edge of the read. Based on these properties, the HyperPlus endonuclease is considered to be an endonuclease(s), which prefer DNA sequences with diverse palindromic structure (over 1,000 palindromes with different lengths and sequences) without any specificity. Since a restriction enzyme is defined as an endonuclease with specific recognition site [19], we speculate that the HyperPlus endonuclease is not a mixture of restriction enzymes. Nevertheless, limited information prevented us from further inferring the exact enzyme(s) responsible for the sequencing noise measured in our study. Other endonucleases for DNA fragmentation, such as Fragmentase [14], may also generate sequencing noise that could be misinterpreted as genuine mutations. Fragmentase is a mix of two enzymes: one randomly creates nicks in the dsDNA while the other one cuts the strand opposite to the nicks. It is possible that the noise created by Fragmentase could be similarly ameliorated from the data through a specific algorithm, like the one employed in this study.

Given that endonucleases themselves are incapable of incorporating nucleotides into the DNA or causing mutations [19], we speculate that mutations arise after enzymatic fragmentation during the "fill-in process" orchestrated by the DNA polymerase for end repair ("End repair & A-tailing enzyme" prior to adaptor ligation in the HyperPlus kit). Ultra-sonication randomly cleaves DNA molecules at different genomic positions and, therefore, in the subsequent fill-in process, nucleotides are incorporated at different genomic positions in different DNA molecules. Even if an erroneous nucleotide is incorporated into the cleaved sites, the resultant artifact would not be recognized as a mutation, because it would not consistently appear at the same position on different molecules. However, because the HyperPlus endonuclease preferentially cleaves specific sites on the DNA, when an erroneous nucleotide is incorporated, the resultant artifact could be mistakenly recognized as a mutation because it appears repeatedly at the same position on different molecules. For instance, hairpin structures made in palindromes may result in nucleotide mis-incorporation into the center of a palindromic sequence, which would ordinarily be detectable as a mutation, albeit incorrectly. Moreover, multi-nucleotide substitutions near the end of the read—another feature of the artifactual noise—can arise as more than one mis-incorporation during the fill-in process. By filtering the data using our algorithm, these positional biases and other artifacts are identified and excluded, thereby minimizing the number of non-genuine mutations. For instance, the algorithm designed in this study will identify and exclude mutation-based sequencing artifacts within the center of palindromic sequences, as well as multi-nucleotide substitutions near the ends of the read.

We found a substantial number of somatic SNVs/indels in the paired analysis of the six tumor samples using the SureSelect treatment for normal samples and the HyperPlus treatment for tumor samples (SH). We considered that such noise could be avoided by using the same DNA fragmentation method for paired samples (i.e., HH combination), and tested this using samples from two rectal cancer cases. Even though we confirmed a substantial reduction in the number of SNVs/indels using just one fragmentation method, upon careful examination, we detected the persistence of HyperPlus noise among the resultant SNVs/indels from the HH combination; this noise was frequently classified by the algorithm in other pairwise comparisons and characterized by palindromic structure. This finding reinforces our proposal of the risk that persistent errors may be confused with genuine mutations due to their recurrent appearance in a cohort. In such situations, the algorithm developed in this study can be

used to distinguish true mutations from sequencing errors. The current study hence provides the technical basis to remove sequencing noise derived from HyperPlus endonuclease treatment.

## Materials and methods

### Starting amount of DNA

In our sequencing facility, the default protocol for library preparation in exome sequencing is the use of the SureSelect kit (Agilent Technologies). In cases where there is less than 200 ng of DNA, we use the HyperPlus (KAPA Biosystems) kit. Thus, for the purposes of this comparative study, the starting amounts of DNA were 40 and 200 ng for preparation with the HyperPlus and SureSelect kits, respectively.

### DNA fragmentation by ultra-sonication

DNA shearing by ultra-sonication was performed with the E220 Focused-ultra-sonicator (Covaris) for 360 s at 4˚C according to the manufacturer's recommendations. After shearing, the median peak in fragment length was 177 bp (range, 160–185 bp), as measured using the 2200 TapeStation (Agilent Technologies).

### DNA fragmentation using HyperPlus endonuclease

DNA was incubated with the HyperPlus "Frag Enzyme" (KAPA Biosystems) at 37˚C for 30 min, according to the manufacturer's recommendations.

### Library preparation

After enzymatic fragmentation (HyperPlus) or ultrasonic shearing (SureSelect), we performed end repair, phosphorylation, and the ligation of barcoded adaptors according to each of the manufacturer's protocols. DNA samples were then captured by hybrid capture using the SureSelect Human All Exon V5 kit (Agilent Technologies). The captured libraries were amplified with the addition of index sequences, and were multiplexed before sequencing.

### Sequencing

Libraries were sequenced using the HiSeq2500 (Illumina), according to the manufacturer's recommendations, with a median depth of coverage of 260 (124–271) per tumor with the HyperPlus kit, 294 (257–334) per tumor with the SureSelect kit, and 172 (148–225) per normal tissue sample with the SureSelect kit.

### Bioinformatics tools for somatic SNV/indel calls

Sequenced reads were aligned with BWA (Burrows-Wheeler Aligner; ver. 0.7.12) to the human genome reference (hg19) [20]. GATK (GenomeAnalysisTK; ver 3.4–46) was used to recalibrate the variant quality score and to perform local realignment [21]. Somatic SNVs were called with VarScan (ver. 2.3.7), MuTect (ver.1.1.5), and Karkinos (ver. 3.0.22) [22–24]. VarScan (ver. 2.3.7), SomaticIndelDetector (ver.2.3–9), and Karkinos2 (ver.0.1) were used to detect somatic indels [21, 22, 24]. SNVs and indels were considered as genuine only when they were detected by at least two of three callers and used for subsequent analyses. SNVs/indels were annotated with ANNOVAR [25] (2015 Mar 22 released version). COSMIC (Catalogue Of Somatic Mutations In Cancer; v82) database [12] was integrated into the ANNOVAR database and used to identify the SNVs/indels registered in the COSMIC database.

## Informatics methods to characterize and filter sequencing artifacts by HyperPlus

We designated a somatic SNV coincidently located at the center of a palindromic sequence as an "SNV-centered palindrome" (SCP). The SNV-centered sequences were extracted from a reference FASTA file (hg19.fa; http://hgdownload.cse.ucsc.edu/goldenPath/hg19/) and were determined to have palindromic structure or not by an in-house script. Reads with and without somatic SNVs/indels were extracted from a BAM file derived from a tumor sample. The position of a variant in a read was subsequently assigned according to the mapped position, and the CIGAR string of the read with an in-house Python (version 3.7.2) script using the Pysam module (https://github.com/pysam-developers/pysam).

COSMIC mutational signatures [9] were computed from trinucleotide frequencies of nucleotide substitutions in each sample. Probabilities for the signatures were downloaded from the COSMIC website (https://cancer.sanger.ac.uk/cosmic/signatures). Each signature contribution to a tumor signature profile was computed by minimizing the difference between the trinucleotide frequencies and the linear sum of probabilities using the Rsolnp (version 1.1.6) library.

Variant recurrence was counted when a variant in a sample had the identical genomic coordinate and altered nucleotides to that in the other samples among the 28 tumors, for which the libraries were prepared with the HyperPlus kit. We utilized this variant recurrence as part of the filtering algorithm for data containing artifacts generated due to preparation with the HyperPlus kit. Because genuine driver mutations were also recurrent in various types of cancer and because most of them were already registered in the COSMIC database, these "COSMIC database-recurrent" variants were returned for further filtering.

To remove SNVs/indels having positional bias in a read, we used $p$-values from Kolmogorov-Smirnov (KS) comparisons of the position between the variant and wildtype alleles in a read with R (version 3.5.2). The ratio of soft-clipped reads per total reads with SNVs/indels was computed according to information from the BAM file, and this was also used for filtering. A predictive model was developed based on the logistic regression to classify the SNV/indel as noise or signal. Threshold lines were estimated by generalized linear models with the logit link function. The threshold line was then optimized to provide the maximum summation of sensitivity and specificity.

## Supporting information

**S1 Fig. Location of SNV-centered palindromes (SCPs) on the read.** Each panel indicates the location of SCPs filtered by the noise-canceling algorithm ($x$-axis) and the number of reads ($y$-axis) for each odd-numbered-length palindrome. Palindrome length is indicated by the number at the right shoulder of each panel. Palindromic sequences and SNVs are shown as black lines and red points, respectively. Thin and thick dashed lines indicate the end and 30 bases from edge of the read, respectively. Each read is 100 bases in length, as shown above the panel. (DOCX)

**S1 Table. Sequence and number of filtered palindrome sequences.** (XLSX)

## Acknowledgments

The authors thank Ken Uchibori, Makoto Nishio, Takashi Akiyoshi, Takeru Wakatsuki, Keisho Chin, Ken Namikawa and Junko Fujisaki for kindly allowing us to re-analyze their data originally generated for other genomic cohort studies. The authors also thank Tetsuo Noda,

Kazuma Kiyotani, Yasuo Uemura, Izuma Nakayama and Yu Imamura for helpful discussions. The authors also thank Megumi Nakai, Tomoko Kaneyasu, Sayuri Amino, Yuki Ota, Rie Furuya and Ikumi Haraguchi, for technical assistance; Minako Hoshida for administrative assistance; and Rebecca Jackson for editing a draft of this manuscript.

## Author Contributions

**Conceptualization:** Seiichi Mori.

**Data curation:** Akihisa Takahara, Taichi Hagio, Osamu Gotoh.

**Investigation:** Norio Tanaka, Akihisa Takahara.

**Methodology:** Norio Tanaka, Akihisa Takahara, Taichi Hagio, Rika Nishiko, Junko Kanayama.

**Project administration:** Seiichi Mori.

**Software:** Norio Tanaka, Akihisa Takahara, Taichi Hagio, Osamu Gotoh.

**Supervision:** Seiichi Mori.

**Visualization:** Norio Tanaka, Akihisa Takahara, Taichi Hagio, Osamu Gotoh.

**Writing – original draft:** Norio Tanaka, Akihisa Takahara, Seiichi Mori.

**Writing – review & editing:** Seiichi Mori.

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
