## [Decision Letter · Decision Letter 0]

31 Oct 2019

PONE-D-19-26447

Sequencing Artifacts Derived from a Library Preparation Method using Enzymatic Fragmentation

PLOS ONE

Dear Dr. Mori,

Thank you for submitting your manuscript to PLOS ONE. After careful consideration, we feel that it has merit but does not fully meet PLOS ONE’s publication criteria as it currently stands. Therefore, we invite you to submit a revised version of the manuscript that addresses the points raised during the review process.

We would appreciate receiving your revised manuscript by Dec 15 2019 11:59PM. To enhance the reproducibility of your results, we recommend that if applicable you deposit your laboratory protocols in protocols.io, where a protocol can be assigned its own identifier (DOI) such that it can be cited independently in the future. For instructions see: http://journals.plos.org/plosone/s/submission-guidelines#loc-laboratory-protocols

We look forward to receiving your revised manuscript.

Kind regards,

Ruslan Kalendar, PhD

Academic Editor

PLOS ONE

Journal Requirements:

MEXT | Japan Society for the Promotion of Science (JSPS) - JP15K06861 [S. M. ]

MEXT | Japan Society for the Promotion of Science (JSPS) - JP17K18337 [O. G. ]

MEXT | Japan Society for the Promotion of Science (JSPS) - JP18K07338 [S. M. ] 

We note that one or more of the authors are employed by a commercial company: Data4C’s Co. Ltd.

Reviewers' comments:

Reviewer's Responses to Questions

**Comments to the Author**

1. Is the manuscript technically sound, and do the data support the conclusions?

Reviewer #1: Yes

Reviewer #2: Yes

2. Has the statistical analysis been performed appropriately and rigorously? 

Reviewer #1: N/A

Reviewer #2: Yes

3. Have the authors made all data underlying the findings in their manuscript fully available?

Reviewer #1: No

Reviewer #2: Yes

4. Is the manuscript presented in an intelligible fashion and written in standard English?

Reviewer #1: Yes

Reviewer #2: Yes

5. Review Comments to the Author

Reviewer #1:

This paper presents an interesting comparison of the effects of preparing DNA fragments using two types of kits - sonication and Restriction enzyme based. The authors have deveoped a way to bring the restriction enzyme fragments into agreement with the sonicated samples. Unfortunately not all of the data set used are publicly available since they are patient data.

The paper does, however, need some additional background information. Some summary of the number of restriction enzymes that act upon palindromic sequences, and a clear explanation of why this leads to the errors that their approach is able to correct. One wonders whether it might be possible to infer which restriction enzymes are actually used in the kits, but this might raise intellectual property issues.

Reviewer #2:

High throughput, short-read DNA sequencing has become an essential tool for the analysis and targeted treatment of human cancers. Samples for sequencing are derived from human tumors (and control samples), and the quantities available for analysis are typically very small. Therefore technologies have been developed to facilitate accurate sequencing from very limited input material. A key step in the preparation of these samples is the fragmentation of the isolated DNA. The standard method for fragmentation has been ultra-sonication. However, ultra-sonication can lead to loss of material. Given the small amount of DNA in the initial samples, this can be a serious problem. Therefore, DNA isolation protocols have recently been introduced using endonucleases as an alternative to ultra-sonication. Unfortunately, however, the sequences obtained from the two preparations (fragmentation by endonucleases versus ultra-sonication) are not identical. Which is correct? And can one develop computational methods to identify and remove the errors?

Addressing these two challenges is the focus of the manuscript by Tanaka et al. The authors sequenced a range of samples from healthy and cancerous tissues using fragmentation either by endonucleases or by ultra-sonication. Using pairwise comparisons of many sequences from both methods, they show that fragmentation by endonucleases introduces errors. Moreover, they show that these errors are typically introduced in regions of DNA close to palindromic sequences.

Building on their understanding of the distinctive features of these errors, the authors developed an algorithm to identify the artifactual noise. They then used this algorithm to cancel this noise and recover correct sequences.

This is an important study that will find use, as more and more scientists and clinicians rely on high throughput sequencing of very small samples. The science is compelling and the manuscript is clear and well written.

This referee does not have any recommendations for revisions except for one small issue. I would like to see more discussion (perhaps a paragraph) about why enzymatic cleavage introduces errors. This finding is at the core of this whole study, and it’s perplexing. I understand the answer is not fully clear. Nonetheless will be good to see a few more sentences speculating about why enzymatic cleavage causes errors.

6. PLOS authors have the option to publish the peer review history of their article (what does this mean?). If published, this will include your full peer review and any attached files.

Reviewer #1: No

Reviewer #2: No

---

## [Author Response · Author response to Decision Letter 0]

4 Dec 2019

Reviewer #1:

This paper presents an interesting comparison of the effects of preparing DNA fragments using two types of kits - sonication and restriction enzyme based. The authors have developed a way to bring the restriction enzyme fragments into agreement with the sonicated samples. Unfortunately not all of the data set used are publicly available since they are patient data.

The paper does, however, need some additional background information. Some summary of the number of restriction enzymes that act upon palindromic sequences, and a clear explanation of why this leads to the errors that their approach is able to correct. One wonders whether it might be possible to infer which restriction enzymes are actually used in the kits, but this might raise intellectual property issues.

This comment is related to the comment raised by Reviewer #2. In the previous version of the manuscript, we neither showed any detailed information regarding SNV-centered palindromes (SCPs) nor fully discussed the reason(s) why endonuclease treatment introduces artifactual mutations. This was simply because we did not know which endonuclease is used in the HyperPlus kit. However, from the comments raised by the two reviewers, we realized that we could still speculate on the mechanisms by which artifactual mutations were being generated based on the properties of the artifacts—i.e., the palindromic structure, positional bias and multi-nucleotide substitutions—even without knowledge of the enzyme.

In the revised manuscript, we provide information on the frequencies (Figure 5D), locations (on the read; Supplementary Figure 1), and nucleotide sequences of the SCPs (Supplementary Table 1) filtered by the noise-cancelling algorithm. The data showed three characteristic features: first, there was substantial diversity in the lengths and sequences of the SCPs (371-655 [median 568] different palindromes per sample; Figure 5D); second, 66.6% of the SCPs were recurrently observed across the samples (Figure 5D), and; third, in almost all (90.4%) of the SCPs, the entire palindromic sequence was nested within 30 bases from the edge of the read (Supplementary Figure 1). Based on these properties, the HyperPlus endonuclease is considered to be an endonuclease(s), which prefer DNA sequences with diverse palindromic structure (over 1,000 palindromes with different lengths and sequences; Supplementary Table 1) without any specificity. Since a restriction enzyme is defined as an endonuclease with specific recognition site (as a note, over 3,000 type II restriction enzymes, each of which recognizes specific palindrome sequence, have been identified) [1], we speculate that the HyperPlus endonuclease is not a mixture of restriction enzymes. Nevertheless, limited information prevented us from further inferring which specific enzyme is used in the kit. 

In response to the second part of the Reviewer’s concern, given that endonucleases themselves are incapable of incorporating nucleotides into the DNA or causing mutations [2], we speculate that mutations likely arise after enzymatic fragmentation during the fill-in process with the DNA polymerase for end repair (“End repair & A-tailing enzyme” prior to adaptor ligation in the HyperPlus kit). Ultra-sonication randomly cleaves DNA molecules at different genomic positions; therefore, in the subsequent fill-in process, nucleotides are incorporated at different genomic positions in different DNA molecules. Even if an erroneous nucleotide is incorporated into the cleaved site, the resultant artifact would not be recognized as a mutation, because it would not repeatedly appear at the same position on different molecules. However, when nucleotide mis-incorporation occurs in a site preferentially cleaved by the HyperPlus endonuclease, the resultant artifact could be mistakenly recognized as a mutation because it appears repeatedly at the same position on different molecules. For instance, hairpin structures made in palindromes may result in nucleotide mis-incorporation into the center of a palindrome, which would ordinarily be detectable as a mutation; albeit, incorrectly. Moreover, multi-nucleotide substitution near the end of a read—another feature of the artifactual noise we see—can arise as more than one mis-incorporation during the fill-in process. By filtering the data using our algorithm, these positional biases and other artifacts are identified and excluded, thereby minimizing the number of non-genuine mutations. For instance, the algorithm designed in this study will identify and exclude mutation-based sequencing artifacts within the center of palindromic sequences, as well as multi-nucleotide substitutions near the ends of the read.

As mentioned above, the revised manuscript includes information on the frequencies (Figure 5D), locations (on the read; Supplementary Figure 1), and nucleotide sequences of the SCPs (Supplementary Table 1) filtered by the noise-cancelling algorithm. We accordingly amended our description in the section “Noise Reduction in the Training Data” in the Results, and include a paragraph in the Discussion to elaborate on these speculations. We hope that these changes address the concern raised by Reviewer 1.

Reviewer #2:

High throughput, short-read DNA sequencing has become an essential tool for the analysis and targeted treatment of human cancers. Samples for sequencing are derived from human tumors (and control samples), and the quantities available for analysis are typically very small. Therefore technologies have been developed to facilitate accurate sequencing from very limited input material. A key step in the preparation of these samples is the fragmentation of the isolated DNA. The standard method for fragmentation has been ultra-sonication. However, ultra-sonication can lead to loss of material. Given the small amount of DNA in the initial samples, this can be a serious problem. Therefore, DNA isolation protocols have recently been introduced using endonucleases as an alternative to ultra-sonication. Unfortunately, however, the sequences obtained from the two preparations (fragmentation by endonucleases versus ultra-sonication) are not identical. Which is correct? And can one develop computational methods to identify and remove the errors?

Addressing these two challenges is the focus of the manuscript by Tanaka et al. The authors sequenced a range of samples from healthy and cancerous tissues using fragmentation either by endonucleases or by ultra-sonication. Using pairwise comparisons of many sequences from both methods, they show that fragmentation by endonucleases introduces errors. Moreover, they show that these errors are typically introduced in regions of DNA close to palindromic sequences.

Building on their understanding of the distinctive features of these errors, the authors developed an algorithm to identify the artifactual noise. They then used this algorithm to cancel this noise and recover correct sequences.

This is an important study that will find use, as more and more scientists and clinicians rely on high throughput sequencing of very small samples. The science is compelling and the manuscript is clear and well written.

This referee does not have any recommendations for revisions except for one small issue. I would like to see more discussion (perhaps a paragraph) about why enzymatic cleavage introduces errors. This finding is at the core of this whole study, and it’s perplexing. I understand the answer is not fully clear. Nonetheless will be good to see a few more sentences speculating about why enzymatic cleavage causes errors.

This comment is related to the comment raised by Reviewer #1. As the Reviewers have pointed out, endonucleases cannot incorporate nucleotides into DNA [2], and, therefore, endonuclease treatment does not directly cause erroneous nucleotide incorporation. We speculate that mutations likely arise as part of the fill-in process by the DNA polymerase for end repair (“End repair & A-tailing enzyme” prior to adaptor ligation in the HyperPlus kit) after enzymatic fragmentation. Because nucleotide mis-incorporation occurs at sites preferentially cleaved by the HyperPlus endonuclease, the resultant artifacts are recognizable because of their recurrence at the same position on different molecules. We added this comment as a paragraph into the Discussion.

Reference

1. Roberts RJ, Vincze T, Posfai J, and Macelis D. REBASE—enzymes and genes for DNA restriction and modification. Nucleic Acids Res. Volume 35:D269–270 (2007)

2. Roberts RJ, Halford SE. Type II Restriction Enzymes. In Roberts RJ, Linn SM, Lloyd RS, editors, Nucleases, 2nd Ed. Cold Spring Harbor Laboratory Press. pp. 35-88 (1993)

---

## [Editor Report · Decision Letter 1]

19 Dec 2019

Sequencing Artifacts Derived from a Library Preparation Method using Enzymatic Fragmentation

PONE-D-19-26447R1

Dear Dr. Mori,

We are pleased to inform you that your manuscript has been judged scientifically suitable for publication and will be formally accepted for publication once it complies with all outstanding technical requirements.

With kind regards,

Ruslan Kalendar, PhD

Academic Editor

PLOS ONE

---

## [Editor Report · Acceptance letter]

26 Dec 2019

PONE-D-19-26447R1 

Sequencing Artifacts Derived from a Library Preparation Method using Enzymatic Fragmentation 

Dear Dr. Mori:

I am pleased to inform you that your manuscript has been deemed suitable for publication in PLOS ONE. Congratulations! Your manuscript is now with our production department. 

With kind regards,

on behalf of

Dr. Ruslan Kalendar 

Academic Editor

PLOS ONE